# Allogeneic Transplantation in Multiple Myeloma—Does It Still Have a Place?

**DOI:** 10.3390/jcm9072180

**Published:** 2020-07-10

**Authors:** Gösta Gahrton, Simona Iacobelli, Laurent Garderet, Ibrahim Yakoub-Agha, Stefan Schönland

**Affiliations:** 1Department of Medicine, Karolinska Institutet, Huddinge, SE 14186 Stockholm, Sweden; 2Department of Biology, University of Rome Tor Vergata, 00133 Rome, Italy; simona.iacobelli@ebmt.org; 3Department of Hematology and Cellular Therapy, Hospital Hospital-Pitié Salpêtrière, 75013 Paris, France; laurent.garderet@aphp.fr; 4CHU de Lille, Université de Lille, INSERM U1286, Infinite, 59000 Lille, France; ibrahim.yakoubagha@chru-lille.fr; 5Department of Internal Medicine V, University of Heidelberg, 69120 Heidelberg, Germany; Stefan.Schoenland@med.uni-heidelberg.de

**Keywords:** allogeneic transplantation, multiple myeloma

## Abstract

Novel drugs have improved survival for patients with multiple myeloma in recent years. However, the disease is still fatal. Allogeneic stem cell transplantation (Allo) has proven to cure some patients with the disease, but its role is controversial due to relatively high transplant-related toxicity and mortality (nonrelapse mortality, NRM). Using nonmyeloablative reduced-intensity conditioning (RIC), both toxicity and NRM can be reduced, and RICAllo is, therefore, an option for subgroups of patients. Upfront tandem autologous/RICAllo (Auto/RICAllo) was shown to be superior to single Auto or tandem Auto/Auto in both progression-free (PFS) and overall survival (OS) in two prospective studies with long-term follow-up, while three similarly designed studies did not detect a difference. A recent update of pooled patient data from four of these studies showed significantly superior PFS and OS with Auto/RICAllo. Importantly, none of these studies showed inferior results with Auto/RICAllo in patients less than 70 years of age. Auto/RICAllo appears to overcome some poor risk cytogenetic markers. Encouraging results have also been seen in treatment of relapsed patients. Combining Allo with new proteasome inhibitors and immunomodulatory drugs may further improve results. Other encouraging new cell therapies such as with CAR T-cells, NK- and CAR NK-cells may well have a place in combination with RICAllo. Such studies are warranted.

## 1. Introduction

The recent progress in the treatment of multiple myeloma (MM) has been impressive. Numerous new drugs have shown to be effective in prolonging survival. For patients up to about 70 years of age, combining the new drugs with autologous (Auto) stem cell transplantation results in median overall survival (OS) of about 6–7 years. However, despite this progress, eventually all patients seem to relapse or progress and, finally, succumb from their disease.

Allogeneic transplantation (Allo) was introduced in an attempt to cure MM, the idea being to take advantage of the dual effect of direct cytotoxicity and an immune graft vs. tumor action. The first Allo in a patient with MM was performed in Seattle by Donnall Thomas group in 1957 as part of a transplant series of six patients with hematologic malignancies [1]. However, all patients died, and only one had signs of engraftment. Few colleagues believed at this time in Allo as a treatment for hematopoietic disorders. However, in the mid-1980s a few other centers presented case reports with encouraging results [2,3,4,5]. A woman with refractory MM received an Allo at the Karolinska University Hospital, Huddinge in 1983 [5]. She entered a complete remission (CR) that lasted for more than 3 years, but subsequently relapsed, and thus, the hope that the first patient with MM had been cured by Allo vanished. However, attempts to use Allo in MM continued and retrospective results on the first larger series of patients were published in 1987 [6] and 1991 [7] by the EBMT (European Society for Blood and Marrow Transplantation). A significant proportion of patients achieved CR, which was shown to be the most important prognostic factor for long-term survival [8]. A serious concern with high-dose myeloablative conditioning, as used in the original Seattle protocol for leukemia [9,10], was a very high treatment-related mortality in MM [8]. A graft-vs.-myeloma (GVM) effect [11,12,13] was well documented, and the idea to rely more on this effect than on the direct cytotoxic effect resulted in the use of nonmyeloablative reduced-intensity conditioning (RIC) regimens in order to reduce transplant-related mortality [14,15,16] 

Nevertheless, Allo in MM is a controversial treatment and many centers claim that it should not be performed at all. However, other centers still perform Allo. Although some decrease in activity can be seen, particularly among the US centers, about 400 Allos are reported to the EBMT registry and some 300 to the CIBMTR (Center for International Blood and Marrow Transplant Research) registry every year.

In this review, results with myeloablative conditioning before Allo will only be briefly summarized, while the focus will instead be on results with reduced-intensity conditioning allogeneic transplants (RICAllo) and how it could be combined with proteasome inhibitors, immunomodulators, antibodies, and cellular immune therapy.

## 2. Myeloablative Conditioning

Myeloablative conditioning regimens before Allo has the aim of eradicating the disease and rescue the patient with the normal stem cells in the allogeneic graft. In addition, advantage is taken of a graft vs. myeloma (GVM) effect documented early [11,12,13]. Myeloablation is most frequently obtained with total body irradiation 10–12 Gy, fractionated or unfractionated with lung shielding [7,8], but drugs like busulfan and melphalan in high dosages are also used [17,18,19]. A lower relapse rate (RR) after Allo as compared to Auto is well documented, but high transplant-related mortality of 30–40%, in part due to severe graft-vs.-host disease (GVHD), hampers both progression-free survival (PFS) and overall survival (OS) [8,20]. However, some patient groups do better, e.g., women transplanted with a female donor had low NRM in one EBMT study, resulting in a 9-year survival of 30% [20]. The highest NRM was seen in female to male transplants, but due to a low relapse rate, survival did not differ from that in male to male transplants [21,22]. The graft source was bone marrow in early transplants, but peripheral blood stem cells are as effective and is now the most common source. They seem to induce somewhat more GVHD, but similar OS as with bone marrow [23]. Due to better supportive treatment, results with myeloablative conditioning improved significantly with time. In a comparison between patients transplanted 1994–1998 and 1983–1993, OS improved from 32% to 50% at 4 years [24], but NRM was still high (30%) at 2 years. 

Myeloablative conditioned transplants induced CR in 50–60% of patients [8,25], and CR was the most important factor predicting long-term survival. Furthermore, higher rates of molecular remission after myeloablative Allo than after Auto [26,27] were documented already at this time. However, despite these encouraging results, the high transplant-related mortality has prevented common use of myeloablative conditioning. The concept of reduced-intensity (nonmyeloablative) conditioning (RICAllo) was instead introduced, e.g., conditioning with irradiation as low as 2 Gy in contrast to myeloablation with 10–12 Gy. However, as seen below, the higher relapse/progression rate with RICAllo transplantation, even with a previous Auto, compared with myeloablative conditioning may be a reason to again try more intensive conditioning in selected high-risk relapsed or refractory patients. 

## 3. Reduced-Intensity (Nonmyeloablative) Conditioning Allogeneic Transplantation (RICAllo) as Upfront Treatment—Prospective Trials

Nonmyeloablative reduced-intensity conditioning (RIC) aims at minimizing the NRM and toxicity and rely more on the GVM for antimyeloma efficacy. The original Seattle Group protocol was based on canine transplant studies [28,29]. It included 2 Gy irradiation for conditioning and GVHD prophylaxis with mycophenolate mofetil and cyclosporine [16]. Encouraging results in the animals paved the way for numerous Phase I and Phase II studies in human MM [16,17,30,31,32,33,34,35,36,37,38,39] and eventually for prospective trials comparing RICAllo and Auto as upfront treatment of MM.

There are seven prospective studies of upfront treatment with somewhat different design and inclusion criteria and one study of pooled patient data from four of these studies. Five of these studies included both standard and high-risk patients (Table 1), while two included only high-risk ones defined by β2-micro ≥ 3 or 4 mg/L and/or presence of chromosome del13. However, since high-risk patients in the two studies that included both standard and high-risk patients could be analyzed separately, there were four studies analyzing outcome in high-risk patients (Table 2). 

Six of the studies were started before availability of the new drugs. Although new drugs may have been used as they developed later in the course of the disease, their impact on the outcome has not been evaluated or reported. All seven studies are based on the availability of an HLA-matched sibling donor [40,41,42,43,45,47,48,50,51]. They all utilize tandem transplants, with the initial Auto being used as debulking and the RICAllo usually done within 6 months from the Auto. 

In six of the studies, the control group was tandem Auto in those patients that lacked a donor, whereas in one of the studies, either single or tandem Auto was used. The induction treatment was vincristine + adriamycin + dexamethasone (VAD) or similar combinations in six studies, and the conditioning for the initial Auto was 200 mg/m^2^ melphalan. 

The first study reporting superior results with the Auto/RICAllo regimen was from the **Italian group** [40,41] (Table 1). It comprised 245 patients enrolled at time of MM diagnosis. Eighty patients out of 162 that were typed for HLA, had an HLA-identical sibling donor, but only 58 completed the Auto/RICAllo transplant. Out of those, who lacked a compatible donor, 46 received the planned Auto/Auto treatment. The study was analyzed repeatedly. In a the latest update, more than 7 years from diagnosis, the median survival was not reached in the Auto/RICAllo group vs. 4.2 years in the Auto/Auto group (*p* = 0 0.001) [41]. The long-term follow-up is a strength in this study, but it is, to some extent, counterbalanced by the weakness of relatively low number of patients that completed the Auto/RIC procedure and the low number of patients in the control Auto/Auto group 

A smaller study was reported by the **PETHEMA** group [42]. It included only 25 patients in the Auto/RICAllo arm compared to 85 receiving Auto/Auto. Only patients less than 70 years who failed to achieve a CR or nCR (near CR) after first Auto were eligible for RICAllo in the trial group and second Auto in the control group. The type of second transplant was based on the availability of an HLA-compatible sibling donor. The conditioning for Allo was melphalan + fludarabine. In the control arm, patients received either CBV (cyclophosphamide, BCNU, etoposide) or high-dose melphalan for the second transplant. Although the difference was not significant, Auto/RICAllo tended to be superior with the median time for PFS and OS did not reach compared to 31 months (*p* = 0.08) and 58 months (*p* = 0.9), respectively, in the Auto/Auto group. Thus, this study indicated a trend towards an advantage with Auto/RICAllo in patients who did not reach CR after first Auto.

The **HOVON-50** study [43,44] included 260-HLA typed patients, of whom, 122 had an identical sibling donor, while 138 did not. Ninety-nine out of the 122 patients with a donor received an Auto/RICAllo, while those without a donor received Auto/Auto or treatment with thalidomide maintenance after first Auto. Analysis on an intention to treat basis revealed no significant difference in PFS or OS between Auto/RICAllo and the control group at 8 or 10 years, despite a lower relapse rate with Auto/RICAllo at 8 years of 55% vs. 77%. When only those patients that had really received a RICAllo transplant (*n* = 99) were compared to those who continued maintenance or received a second Auto (*n* = 122), there was a significant advantage in PFS for the RICAllo patients at 8 years. However, the superior PFS in those 99 patients did not translate into a significantly better OS. There was no information about later treatment with novel drugs, except for the treatment with thalidomide.

The largest multicenter prospective trial is the **BMT-CTN 0102** study [45,46]. It included 710 patients < 70 years of age, of whom, 625 had standard risk disease. One hundred and fifty-six (83%) of 189 patients with standard risk disease received Auto/RICAllo based on the availability of an HLA-identical sibling donor and 366 (84%) of 436 without a donor received Auto/Auto. Auto conditioning was high-dose melphalan (200 mg/m^2^), and RIC was single dose 2 Gy total body irradiation. Primary endpoints in patients with standard risk myeloma, defined as β2 microglobulin < 3.0 mg/L and absence of deletion 13 by classic karyotyping, was PFS and OS. In the original report, there was no significant difference in the 3-year PFS or OS between the Auto/Auto and Auto/RICAllo. In a recent update [53], there was still no significant difference between PFS and OS at 6 and 10 years, respectively, in standard risk patients in contrast to results in high-risk patients (see below). 

The second largest study, the EBMT study [47,48], comprised 357 MM patients recruited from 23 European centers. Patients < 70 years of age with an HLA-identical sibling were allocated to Auto/RICAllo (*n* = 108) and those without to Auto or Auto/Auto (*n* = 249). The study was first published in 2011 [47] and updated in 2013 [48] with a median follow-up of 96 months. PFS and OS were significantly superior with Auto/RICAllo as compared to the Auto or Auto/Auto at this time, i.e., 22% vs. 12% (*p* = 0.027) and 49% vs. 36% (*p* = 0.030) at 96 months, respectively. The reason for the superior PFS and OS with Auto/RICAllo was a lower relapse rate despite a higher NRM with Auto/RICAllo. It is important to note that at 36 months, there was no significant difference in PFS or OS in Auto/RICAllo vs. Auto or Auto/Auto, emphasizing that long-term follow-up is necessary to see the benefits of Auto/RICAllo. When analyzing only those patients who had received the Auto/RICAllo transplant (*n* = 92) as well as Auto/Auto (*n* = 104) according to protocol, the differences were similar. An additional finding was that CR was the important factor for long-term PFS, irrespective of treatment modality, but a CR obtained with Auto/RICAllo was better sustainable than obtained with Auto/Auto (*p* = 0.027) [54].

This study has recently been updated (Gahrton et al. 2019 unpublished). The 10-year survival is now 47.6% in the Auto/RICAllo group and 27.1% in the Auto group (*p* = 0.0018). If the comparison is made between those who received the second transplant, the results are similar 47.0% vs. 26.6% at 10 years from second transplant, respectively (*p* = 0.0113). Of interest is also that out of 53 patients in the Auto/RICAllo group and 173 in the Auto group, who have relapsed, OS from relapse was superior in the Auto/RICAllo group, i.e. 28.4% and 14.7% at 10 years in the Auto/RICAllo and Auto groups, respectively, irrespective of later treatments. 

## 4. Pooled Data Analysis of the Torino, PETHEMA, EBMT, and BMT-CTN Studies

Recently a pooled analysis was performed of individual patient data from the Italian-Torino (*n* = 162), Spanish PETHEMA (*n* = 110), EBMT—NMAM2000 (*n* = 357), and BMT-CTN (*n* = 709) studies [49]. There were 1338 patients included, 439 in the Auto/RICAllo group and 899 in the Auto/Auto group. In this updated analysis, the median follow-up of survivors was 118.5 months. Overall survival was 62.3% vs. 59.8% at 5 years and 44.1% vs. 36.4% at 10 years (*p* = 0.01) for Auto/RICAllo and Auto/Auto, respectively.

PFS was also improved in Auto/RICAllo (*p* = 0.004) with 5-year PFS of 30.1% vs. 23.4% (*p* = 0.01) and 10-year PFS of 18.7% vs. 14.4% (*p* = 0.06). Risk of NRM was higher in Auto/RICAllo (for 10 year, 19.7% vs. 8.3%, *p* = 0.001), while risk of disease progression was higher in Auto/Auto (for 10 year, 77.2% vs. 61.6%, *p* = 0.001) 

There were 685 progressions in Auto/Auto and 266 in Auto/RICAllo. As in the Torino and the EBMT-NMAM2000 studies, the median postrelapse OS was better in Auto/RICAllo group at 62.3 months vs. 41.5 months in Auto/Auto (*p* < 0.001). At 5 years postrelapse, 51.1% in Auto/RICAllo vs. 37.0% in Auto/Auto were alive (*p* < 0.001). 

Thus, including the updated patients of the largest study, the BMT-CTN, this 4-study pooled patients data analysis shows significantly superior PFS and OS with tandem Auto/RICAllo compared to Auto/Auto.

## 5. Upfront Transplantation of High-Risk Patients

Data are scarce concerning upfront allogeneic transplants in high-risk patients. The first study (the Intergroupe Francophone du Myélome study published in 2006, Table 2) comparing Auto/Auto and Auto/RICAllo included only high-risk patients, defined by beta-2 microglobulin of more than 3 mg/L, and deletion of chromosome 13. Based on the availability of an HLA compatible sibling donor, 65 patients were treated with Auto/RICAllo and 219 with tandem Auto (Auto/Auto). Analysis was made on an intention to treat basis and showed trend for inferior OS in the Auto/RICAllo group (*p* = 0.07). The median event-free survival was 19 vs. 22 months and OS of 34 vs. 48 months in the Auto/RICAllo and Auto/Auto groups, respectively. If only comparing patients that underwent the transplant (36 Auto/RICAllo vs. 166 Auto/Auto patients), the outcome was similar, i.e., OS tended to be better with the Auto/Auto (median 57 vs. 41 months) (*p* = 0.08). In a 2008 update, there was still no significant difference in outcome between the two groups of patients. There was no information about use of new drugs in the course of the disease. However, in contrast to the other prospective studies, the GVHD regimen utilized antithymocyte globulin–Imtix Genzyme (2.5 mg/kg/day during 5 days) and busulfan and fludarabine for conditioning. This might have contributed to the trend for poorer outcome with Auto/RICAllo in contrast to all other upfront studies.

In the updated BMT-CTN study, a subanalysis was also done on 85 high-risk patients defined as those having either del13 by conventional chromosome analysis or beta-2 microglobulin ≥ 4 mg/L [46]. At this long-term follow-up, RR (relapse rate) was significantly lower with Auto/RICAllo both at 6 and 10 years. This, in turn, resulted in better PFS with Auto/RICAllo than with Auto/Auto at both 6 and 10 years, 31% vs. 13% and 21% vs. 4%, respectively (Table 2). At 10 years, OS was 37% with Auto/RICAllo vs. 29% with Auto/Auto, but this difference was not statistically significant. Importantly, the NRM was similar to that in standard risk patients. 

In the NMAM2000 study, cytogenetic analysis with respect to chromosome 13 deletion (del13q14) was performed in 214 patients [48,55]. For patients with del13 (*n* = 92), PFS at 60 months was 31% in the Auto/RICAllo group (*n* = 29) and 10% in the Auto (and Auto/Auto) group (*n* = 63) (*p* = 016). OS was 69% in the Auto/RICAllo group and 52% in the Auto group. At 96 months, corresponding PFS and OS were 21% vs. 5% and 47% vs. 31%, respectively. For patients who were negative for del13, PFS at 60 months was 44% in the Auto/RICAllo group and 20% (*p* = 0.017) in the Auto group, whereas OS was 70% and 61% (*p* = 0.363), respectively. Thus, the outcome tended to be better with Auto/RICAllo in del13 patients and, thus, tended to overcome the poor prognostic impact seen with other treatments.

In the pooled data analysis, there were 214 patients defined as high-risk (89 Auto/RICAllo and 125 Auto/Auto) [49]. High-risk was defined as β2 microglobulin > 4 mg/L or presence of chromosome del13. The 5- and 10-year PFS with Auto/RICAllo was significantly better than with Auto/Auto (32% vs. 17% and 22% vs. 9%, respectively) but a tendency for better OS was not significant (52% vs. 51% and 39% vs. 29%, respectively).

A recent prospective trial of 225 patients with the del13 aberration, 135 patients were allocated to Auto/RICAllo based on the availability of an HLA-matched donor, whereas 90 were allocated to Auto/Auto [52]. The median follow-up was after 91 months. On an intention to treat based analysis, the Auto/RICAllo group had a significantly better median PFS as compared to Auto/Auto, 34.5 vs. 21.8 months (*p* = 0.003), however, with no significant difference in OS. In contrast, in patients with both del13 and del17p, PFS as well as OS was superior with Auto/RICAllo (*n* = 19) compared to Auto/Auto (*n* = 6) (median 37.5 vs. 6.5 months and 61.5 vs. 23.4 months, respectively (*p* = 0.0002 and 0.032)). These results corroborate with those from the NMAM2000 indicating long-term benefits with Auto/RICAllo in patients with del13 but show as well that this is more pronounced if combined with del17p.

## 6. Allogeneic Transplantation as Relapse/Progression Treatment

Allogeneic transplantation in relapsed/progressive patients is more common than upfront transplants [56]. As in the upfront transplants, RIC is the dominating modality for conditioning. Most of the earlier studies include few heavily pretreated patients, use a great variety of conditioning regimens, and include both unrelated and related donors [57,58]. NRM is in the order of 17–25%, and at best PFS and OS reach 26% and 34% at 7 years [59].

Later studies show similar or better results (Table 3).

Schneidawind et al. [61] showed 51% 3-year survival in 41 patients, 30 of them were refractory and 11 were relapsed, which was superior if post-transplant maintenance was used (*n* = 20), increasing OS at 3 years to 68%. 

Patriarca et al. compared 79 patients Allo transplanted in first relapse to 90 patients without a donor that were instead treated with bortezomid + immunmodulators. The transplant group had significantly better 7-year PFS and OS of 9% and 31% compared to 0% and 18%, respectively [63].

Sobh et al. found no significant difference in PFS or OS in relapsed/progressive patients who received a transplant from a 10/10 antigen-matched unrelated donor (MUD) as compared to a 9/10 mismatched one (MMUD); the 5-year PFS and OS were 27% and 39% compared to 14% and 33%, respectively [62]. However, cord blood transplants were inferior, with PFS and OS of 4% and 25% at this time.

In a recently reported retrospective EBMT study [68], 344 patients, 40–60 years of age, that had relapsed after a previous Auto were investigated. The allotransplant was performed between 1991 and 2012. Four groups were compared, i.e., those who were conditioned with MAC (myeloablative conditioning), RIC, nonmyeloablative modalities (NMA), or treated with Auto/RICAllo. There was no significant difference in OS between the nonmyeloablative or RIC approaches, but MAC was significantly worse and only 19.1% survived at 5 years compared to 44.7%, 29%, and 34.2% in NMA, RIC, and Auto/RICAllo, respectively. The poor outcome with MAC appeared to be due to high NRM of 33.2% at 1 year. 

A retrospective study of 96 patients receiving haploidentical transplants as salvage treatment between 2008 and 2016 and reported to the EBMT and CIBMTR registries showed amazingly good results [67]. All patients had received 1–3 previous Autos. The NRM at 1 year was 21%, PFS and OS at 2 years were 17% and 48%, respectively. Although the follow-up was short, the results seemed to be similar to those with matched unrelated donors.

Overall, studies of Allo in relapsed or refractory patients are limited by lack of prospective comparison between Allo and Auto or treatment with new drugs. Also, the number of patients included is in most studies are relatively small. Still, overall, the 3-year OS from the time of salvage treatment is around 50%, and the 5-year OS between 30% and 40% in most later studies [46,61,62,63,64,65]. Maintenance treatment may improve the results [61].

At a consensus meeting including members from the American Society of Blood and Marrow Transplantation (ABMT), European Society of Blood and Marrow Transplantation (EBMT), Blood and Marrow Transplant Clinical Trials Network, and International Myeloma Working Group (IMWG), it was agreed concerning treatment of refractory or relapsed MM [69] that (1) Allo should be considered appropriate therapy for any eligible patient with early relapse (less than 24 months) after primary therapy that included an Auto and/or high-risk features (i.e., cytogenetics, extramedullary disease, plasma cell leukemia, or high lactate dehydrogenase); (2) Allo should be performed in the context of a clinical trial if possible; (3) the role of post-Allo maintenance therapy needs to be explored in the context of well-designed prospective trials; and (4) prospective randomized trials need to be performed to define the role of salvage Allo in patients with MM relapsing after primary therapy. 

## 7. Alternative RIC Regimens

Reduced-intensity conditioning, introduced by the Seattle group [14], was important to reduce NRM in MM, but RR increased in comparison to myeloablative conditioning [23]. However, although higher than with myeloablative conditioning, RR was lower long term than with Auto in all the four largest prospective studies, the BMT-CTN [45,46], the NMAM2000 [47,48], the Italian [40,41], the HOVON [43,69] studies, as well as in the pooled analysis [70]. The conditioning regimen was similar in these studies, i.e., TBI 2 Gy with or without fludarabine. Thus, no conclusion concerning the impact of the conditioning for outcome differences could be made based on these studies. 

The PETHEMA and the IFM used other conditioning regimens, i.e., the PETHEMA used fludarabine 25 mg/m^2^ for 5 days and melphalan 70 mg/m^2^ for 2 days and the IFM used low-dose busulfan combined with high doses of ATG and fludarabine. The poor outcome of Auto/RICAllo in the IFM study may be due to the high ATG dose. A poorer outcome in ATG-conditioned patients has previously been indicated in an EBMT registry study [17,71]. 

Since prospective studies of the impact of the conditioning regimen are lacking, attempts have been made to retrospectively analyze more recently used conditioning regimens. Initial studies using the prodrug Treosulfan (Treo) for conditioning indicated low NRM in MM [72,73]. The treosulfan dose scheduled in the study by Nahi et al. was 14 g/m^2^ × 3, and it was combined with fludarabin 30 mg/m^2^ × 5. Patients transplanted with unrelated donors also received ATG in low dosage. Based on these initial studies, a large retrospective EBMT study compared MM patients conditioned with regimens including treosulfan (Treo) from 2008 to 2015 to those conditioned with other RIC or myeloablative (MAC) regimens [74]. Out of 508 patients who had received Treo, 136 received the transplant upfront either as single Allo (*n* = 38) or tandem Auto/RICAllo (*n* = 98). These 136 patients were compared to 587 upfront patients treated with other RIC (non-Treo/RIC). The 5-year NRM with Treo was 10%, compared to 17% with non-Treo/RIC and 19% with MAC. RR was 59%, 50%, and 49%, respectively, resulting in OS of 62%, 57% and 47% (*p* = 0.04) in Treo, non-Treo/RIC, and MAC at 5 years, respectively. Thus, NRM could be reduced resulting in improved OS that was better than with either non-Treo/RIC or MAC. A similar comparison was performed in later lines of treatment, but here, no significant difference to the advantage of Treo could be documented.

In summary, the conditioning regimen can be improved and is important for outcome. 

## 8. Consolidation and Maintenance 

Little is known about the efficacy of consolidation and/or maintenance treatment following Allo. Thalidomide was used as maintenance in both the HOVON and the BMT-CTN studies, but no significant effect could be seen in any of the two trials. 

Bortezomib appears to have a GVHD preventive effect, while, presumably, still preserving the GVM effect [75,76,77]. Its additional antimyeloma effect makes it an obvious candidate for use in association with allotransplantation. In a recent study of 39 patients (high-risk cytogenetics in 65%) treated with tandem Auto/RICAllo, bortezomib was used both for induction before Auto and as maintenance 1.3 mg/m^2^ for every 2 weeks for 1 year following RICAllo. At 2 years, NRM, PFS, and OS were 6%, 46%, and 92%, respectively [78]. MRD negativity was obtained in 61% of the patients, and RR was only 27% at 2 years, despite the mainly cytogenetic high-risk characteristics. In another recently published EMN (European Myeloma Network) study, 24 patients in second (*n* = 2) or later lines (*n* = 22) of therapy and after previous Auto received bortezomib 1.3 mg/m^2^ during induction/conditioning (days -9 and -2), and, as GVHD prophylaxis, together with tacrolimus and methotrexate. The regimen after day 100 continued as maintenance and included 3-day cycles every 56 day 4–6 times together with lenalidomide (15 mg on day + 1 to + 21), which continued until progression. The regimen resulted in a 4-year EFS and OS of 39% and 72%, respectively, in these mainly high-risk patients that had received multiple previous lines of treatment [79]. 

Lenalidomide has also been used for maintenance following allogeneic transplantation [80,81,82], but more frequently for treatment of relapse following RICAllo. Its use as maintenance is controversial because of induction of GVHD. In the HOVON 79 study [80], 47% of the patients had to stop treatment after 2 cycles due to development of acute GVHD. In the study by Wolschke et al. [81], GVHD was the main reason to discontinue treatment in 29% of the patients. In a study by Becker et al. [82], lenalidomide was used as maintenance in high-risk myeloma. Patients started lenalidomide at a dose of 10 mg at a median of 96 days post-transplant. Tolerability was better than in the HOVON trial with 34% completing treatment but acute GVHD remained an issue with 37% of patients discontinuing therapy due to GVHD. One-year PFS was 68%, which was compelling in this high-risk group.

Thus, maintenance should probably be used following Allo; however, since comparative prospective trials are lacking, firm recommendation are not possible to give. Lenalidomide is probably an option but must start more than 3–5 months following transplant to diminish the risk of severe acute GVHD. 

## 9. Treatment of Relapse Following Allogeneic Transplantation

An interesting observation in the prospective Italian study [41] and the EBMT-NMAM2000 [47,48] was that the postrelapse/progression survival was longer in the Auto/RICAllo group than in the Auto/Auto group. The reason is not fully understood, but the original hypothesis was that some GVM effect [11,12] may persist even after relapse which made new treatment more effective as compared to treatment of relapse in the Auto/Auto arm. The Italian study was later updated, and it was shown that treatment with DLI (donor lymphocytes infusions) postrelapse was an important contributor to the better outcome with Auto/RICAllo, but outcome was also better if only new drugs like lenalidomide had been used, i.e., this treatment appeared to be more effective in treating relapse following Auto/RICAllo than following Auto/Auto [83]. These studies corroborate with studies by Coman et al. [84] showing impressive response rates and a significant association with the development of GVHD after lenalidomide treatment of relapse following allotransplantation as well as by Bensinger et al. [85] who showed that lenalidomide is highly effective in treating relapse/progression after Allo and may be effective without severe GVHD if started later than 1 year after transplantation. Thus, another value of lenalidomide seems to be in treating relapse that occurs late after allotransplantation. These results are in line with a recent large study of patients reported to the CIBMTR registry [86]. Out of 1679 patients that had received an upfront Auto/Auto (*n* = 1186) or Auto/Allo (*n* = 569), 404 and 178 had relapsed, respectively. The postrelapse survival at 6 years was 35% and 44%, respectively (*p* = 0.05). This difference was very similar to the one in the recently updated NMAM2000 study, which was 26.6% and 56.1% at 5 years and 14.7% and 28.4% at 10 years (*p* = 0.0052), respectively (Gahrton et al. unpublished).

A recent study of 60 patients that had relapsed after Allo tried to find predictors for long-term survival [87]. The patients were part of a cohort of 137 allogeneic transplant recipients from 2002 to 2017 who had relapsed after a median follow-up of 4.3 years. The 5-year PFS and OS were 39.3% and 59.5%, respectively. After a median follow up of 2.2 years from the time of relapse, the median postrelapse survival was 1.8 years and 44% survived at 3 years following relapse. Before relapse, the majority of patients had full donor chimerism; thus, deficient chimerism after the transplant was not the reason for relapse, rather a successive loss of GVM might apply, corroborating with previous findings by Rasche et al. [88]. Multivariate analysis demonstrated that high-risk cytogenetics, a short time from transplant to relapse, as previously shown by Franssen et al. [89], and occurrence of acute GVHD before relapse adversely affected postrelapse survival. Unfortunately, the heterogeneity in postrelapse treatment gave little information as to the best option. Patients received a median of 3 lines of therapy beyond relapse/progression. Most of them (80%) received IMiDs (lenalidomide, pomalidomide, and thalidomide), often combined with proteasome inhibitors (74%). Other drugs used were daratumumab (50%), elutuzumab (7%), and check point inhibitors, i.e., nivolumab or pembrolizumab (14%) and 35% received DLI. It appeared that treatment with IMiDs as salvage frequently was associated with GVHD development, contrary to DLI that, in this study, was not associated with GVHD development or worsening of GVHD. 

As in the EBMT-NMAM2000 study, in addition to drugs, some patients that had relapsed following Allo received a second allotransplant or an autologous transplant at relapse. Although in some cases long-term survival was seen, too few patients could be evaluated to make conclusions.

Thus, although data are limited, long-term survival following relapse after Allo appears to be more common than following relapse after Auto. The reason may be a persisting GVM effect that could perhaps be enhanced by DLI with or without the assistance of drugs, particularly immune modulators. 

## 10. Chromosomal Aberrations and Other Prognostic Factors

Certain chromosomal aberrations indicate poor prognosis in MM treated with Auto as well as with drugs [90], in particular, the del17p [91], t (4; 14) [92], t (14; 16), gain (1q) [91], and del8p [93]. The del13q, in some studies, is also a poor prognostic factor, but probably only as a surrogate marker for other poor prognostic aberrations. Allo appears to partly overcome this poor prognostic implication.

In the EBMT–NMAM2000 study (Table 2), patients treated with Auto did worse with than without the del13q aberration but with tandem Auto/RICAllo, there was no significant difference between patients with or without this aberration, and the Auto/RICAllo was superior to Auto in patients with the del13q [47]. 

In a study by Kröger et al. [94], of 73 patients who received Auto/RICAllo, 16 had del17p13 and/or t (4; 14). The 5-year PFS was similar in those with and without the aberrations (24% and 30%. respectively, *p* = 0.70) indicating that the Auto/RICAllo treatment modality could overcome the poor prognostic impact after Auto.

This was confirmed in the prospective study by Knop et al., who showed superior outcome with Auto/RICAllo in patients with both del13q and del 17 [52]. In patients with both aberration the median PFS and OS were 37.5 months and 61.5 months with Auto/RICAllo vs. 6.1 months and 23.4 months with tandem Auto, respectively (*p* = 0.0002 and 0.032).

Thus, it seems that the Auto/RICAllo approach may overcome the poor prognostic impact of the most severe chromosomal aberrations. Still, the number of patients investigated is small, and further studies are needed.

Donor type remains an important prognostic factor [21]. Sibling donors are usually considered to be better than unrelated donors, but with optimal HLA typing and 10/10 antigen compatibility between donor and recipient, there seems to be no significant difference in outcome. In a recent study of patients treated with Allo as salvage after one or two Autos, a 9/10 antigen mismatch transplants seem to be at least as good as a 10/10 ones [62]. Twin donors are the best donors [95,96]. Minor histocompatibility antigens may play a role for the importance of the donor–recipient sex combination. Transplant-related mortality is highest in female to male transplants mainly due to more severe GVHD. On the other hand, the combination involves a stronger GVM effect and lower progression rate. This appeared to be the reason for the lack of difference in OS by using male or female donors for male recipients in the EBMT study [21,22]. Female to female transplants were superior to all other combinations. 

Natural killer (NK) cell function is determined by activating and inhibiting receptors. Killer cell immunoglobulin-like receptor (KIRs) are receptors on NK cells that could be divided in two subgroups A and B–B having more activating receptor genes. In the study by Kröger et al. [97], the outcome for 118 patients was related to donor KIR haplotypes. The best PFS and OS was obtained in the combination male donor with KIR haplotype Bx and the worst with a female donor with KIR haplotype AA.

The CMV (cytomegalovirus) status in donor and recipient has importance by influencing later with risk of severe CMV infection. In an EBMT retrospective study of 413 patients that had received a RICAllo following relapse or progression after Auto [98], the best outcome occurred in CMV-negative patients with a CMV-negative donor. Using a CMV-negative donor for a CMV-positive patient involved no benefit.

Other prognostic factors, such as patient age, stage of disease, and response status, in particular, minimal residual status before and after Allo, seem to be of similar importance as in Auto. 

In summary, prognostic factors are of great importance for selection of patients to allogeneic transplantation. In addition to age, frailty, and response status, the type of chromosomal abnormalities is of crucial importance, and in addition, donor selection remains an important factor for outcome.

## 11. Donor Lymphocyte Transfusions 

Donor lymphocytes infusions (DLI) are used either for treatment of relapse/progression or for improving the quality of response. When used for treatment of relapse/progression after Allo, DLI is effective in 30–60% of patients with responses lasting many years and may, in some cases, be curative [99,100,101,102]. DLI was used in a European multicenter study of 63 patients with refractory or relapsed disease after RICAllo [103]. Twenty-four patients responded; however, 12 showed CR. 

DLI has also been used to prevent relapse following Allo. In a recent study, 61 patients who were not in relapse or progression were treated [104]. An upgrade of the remission status was seen in 54%. Out of these, 41 patients (67%) entered and maintained hematological CR and 26% achieved molecular remission. The 8-year PFS and OS were 43% and 67%, respectively, and the best outcome was seen in those patients who received molecular remission, i.e., PFS at 8 years was 62% and OS was 83%.

DLI is associated with both acute and chronic GVHD. In the study by Gröger et al. [104], acute GVHD grade II–III was seen in 13% (grade III 5%), none had grade IV, and chronic GVHD was mild in 13.1% and moderate/severe in 8.2%. None of the 61 patients treated succumbed from GVHD. DLI associated with chronic GVHD, in another study, seemed to improve PFS and OS [76]. 

Thus, DLI is effective in treatment of relapse and progression, and may also improve response after Allo, but the risk of inducing severe GVHD has to be considered.

## 12. Plasma Cell Leukemia

Primary plasma cell leukemia (pPCL) differs from MM in having > 20% malignant plasma cells in peripheral blood [105]. The prognosis is poor in comparison to MM irrespective of treatment. In a large retrospective study by the EBMT [106] comparing outcome of 272 pPCL and 20,844 MM patients that had received an upfront Auto, the median OS was 25.7 and 62.3 months, respectively. The reason for poor outcome in pPCL was both high NRM and high RR. Allo has been tried by many centers but early results by the CIBMTR were disappointing [107]. A retrospective register study showed poorer outcome for patients treated with Allo than with Auto. Out of 147 pPCL patients, 97 received Auto, whereas 50 received Allo. Despite a lower RR with Allo of 38% vs. 64% with Auto at 3 years, the PFS and OS were poorer in Allo due to a higher NRM of 41% vs. 5%, respectively. However, 34 of the Allo patients received myeloablative conditioning and only 16 received nonmyeloablative (NMA) or RIC, which may have contributed to the high NRM and, in turn, to the poor PFS and OS. 

In a more recent study by the EBMT, 751 patients with pPCL who underwent transplantation during 1998–2014 were investigated [108]. Of these, 70 had an upfront Allo and 681 an upfront Auto. Out of these 681 patients, 122 had a tandem Auto/Auto and 117 had tandem Auto/RICAllo. As in the CIBMTR study, RR was lower with Auto/RICAllo, whereas NRM was higher but the Auto/RICAllo resulted in better PFS and a tendency for better long-term OS in the EBMT study. NRM was only 20% at 2 years as compared to the overall NRM with Allo of 41% at 3 years in the CIBMTR study. 

Thus, the use of tandem Auto/RICAllo as opposed to myeloablative Allo, resulting in lower NRM, seems to be the main reason for the better outcome in the EBMT study. Allotransplantation in this setting, therefore, seems to be a reasonable treatment approach upfront for younger patients with pPCL. 

## 13. Conclusions Concerning Use of Allogeneic Transplantation in Relation to Other Treatments

Allo has the potential to cure some patients with MM so far in contrast to treatment with drugs with or without Auto [109,110]. Based on the prospective trials described in this review, Allo has a place for upfront treatment of fit patients, up to the age of 65–70 years with certain high-risk factors, i.e., the chromosomal aberration del13 + del17 and possibly other ones. Preferably, such treatment should be made in the context of clinical trials. 

Early relapse, within 24 months, is associated with very poor prognosis following drug treatment with or without Auto and should, therefore, be considered for Allo 

Nonmyeloablative reduced-intensity conditioning (RICAllo) should, in general, be the choice of conditioning based on its significantly lower toxicity and nonrelapse mortality (NRM) in the range of 12–15% in upfront treatment. MAC is not recommended due to its higher NRM. 

RICAllo should, in general, be combined with debulking Auto as a tandem Auto/RICAllo which has shown better outcome than Auto/Auto in two out of five prospective studies and in one large study of 1338 patients pooled from four prospective ones.

New conditioning regimens like Treosulfan seem to improve results particularly in upfront Auto/RICAllo, but other RIC regimens could also be considered. Maintenances and intensification treatment with novel drugs should probably be used, although prospective trials in Allo are lacking.

In comparison to many recent studies of novel drugs, with or without Auto in younger patient groups, RICAllo appears to have an advantage mainly in high-risk and relapsed patients, maybe to some extent supported by the use of DLI.

Based on the EBMT study, Allo is an option upfront in younger fit patients with pPCL. 

Survival after relapse following Allo is superior to relapse following Auto, probably due to efficient DLI or utilization of a persistent GVM that can be activated by some drugs, particularly immunomodulating ones. The use of DLI with or without combination with novel drugs could, therefore, be considered both as prevention and for treatments of relapse after Allo.

Allogeneic transplantation might as well be an ideal complement for sustaining response after treatment with new cell therapy, i.e., CAR T-cell and CAR NK-cell treatment. Studies addressing this issue are warranted.

## Figures and Tables

**Table 1 jcm-09-02180-t001:** Prospective studies comparing autologous (Auto) combined with reduced-intensity conditioning (RIC) allogeneic transplantation (Auto/RICAllo) to single Auto or tandem Auto (Auto/Auto) in upfront-treated standard or mixed risk multiple myeloma (MM) patients.

Reference	Conditioning for RICAllo	Transplant Procedure Number of Auto/RICAllo vs. Auto/(Auto) patients (or Donor Available)	NRM Auto/RICAllo vs. Auto/Auto Percent (at year) (y)	Relapse Rate Auto/RICAllo vs. Auto/Auto Percent (at Year) (y)	PFS Auto/RICAllo vs. Auto/(Auto) Median Months (m) or Percent at Year (y)	OS Auto/RICAllo vs. Auto/(Auto) Median Months (m) or Percent at Year (y)
Bruno et al. 2007 [40] Giaccone 2011 [41] (Italian)	TBI 2 Gy	80 vs. 82	16% vs. 2% at 6.5 y	NR	35 vs. 29 m *p* = 0.02	Median 80 m vs. 54 m *p* = 0.01
Rosinol et al. 2008 [42] (PETHEMA)	Flu 25 mg/m^2^ × 5 Mel 70 mg/m^2^ × 2	25 vs. 85	16% vs. 5%	NR	Median not reached vs. 31 m *p* = 0.08	Median not reached vs. 58 m *p* = 0.9
Lokhorst et al. 2012 [43] and 2015 [44] (HOVON)	TBI 2 Gy	Donor vs. No donor 122 vs. 138 Auto/RICAllo vs. Auto/Auto 99 vs. 112	16% vs. 3% 8 y	55% vs. 77% 8 y *p* = 0.001	25% vs. 18% 8 y 27% vs. 15% 8 y	42% vs. 33% 10 y 42% vs. 29% 10 y
Krishnan et al. 2011 [45] Giralt et al 2019 [46] (BMT-CTN)	TBI 2 Gy	189 vs. 436 standard	20% vs. 9% 6 y 20% vs. 11% 10 y	59% vs. 66% 6 y 62% vs. 70% 10 y	22% vs. 25% 6 y 18% vs. 19% 10 y	59% vs. 60% 6 y 44% vs. 43% 10 y
Björkstrand et al. 2011 [47] Gahrton et al. 2013 [48] Gahrton (unpublished) 2019 (NMAM2000)	TBI 2 Gy Flu 30 mg/m^2^ × 3	108 vs. 249	13% vs. 3% at 3 y	60% vs. 82% at 8 y *p* = 0.0002	43% vs. 39% 3 y 22% vs. 12% 8 y *p* = 0.027	75% vs. 68% 3 y 49% vs. 36% 8 y *p* = 0.030 48% vs. 27% 10 y *p* = 0.0018
Costa L 2020 [49] Pooled data from Italian PETHEMA EBMT BMT-CTN	TBI 2 Gy and other	439 vs. 899 All patients	17% vs. 7% 5 y 20% vs. 8% 10 y *p* = 0.001	52% vs. 70% 5 y 62% vs. 77% 10 y *p* = 0.001	30% vs. 23% 5 y 19% vs. 14% 10 y *p* = 0.06	62% vs. 60% 5 y 44% vs. 36% 10 y *p* = 0.01

Auto, autologous; RIC, reduced-intensity conditioning; Auto/RICAllo, autologous/reduced-intensity conditioning allogeneic transplants; MM, multiple myeloma; NRM, nonrelapse mortality; PFS, progression-free survival; OS, overall survival; NR, not reported; Flu, fludarabine; TBI, total body irradiation; Gy, Gray; Mel, melphalan; PETHEMA, Programa Español de Tratamientos en Hematología; HOVON, Hemato-Oncology Foundation for Adults in the Netherlands; BMT-CTN, Blood and Marrow Transplant Clinical Trials Network.

**Table 2 jcm-09-02180-t002:** Prospective studies comparing autologous (Auto) combined with reduced-intensity conditioning (RIC) allogeneic transplantation (Auto/RICAllo) to single Auto or tandem Auto (Auto/Auto) in upfront-treated high-risk MM patients.

Reference	Conditioning for RICAllo	Number of Auto/RICAllo vs. Auto/(Auto) Patients (or Donor Available) High-Risk Criteria	NRM Auto/RICAllo vs. Auto/Auto Percent (at Year) (y)	Relapse rate Auto/RICAllo vs. Auto/Auto Percent (at Year) (y)	PFS Auto/RICAllo vs. Auto/(Auto) Median Months (m) or Percent at Year (y)	OS Auto/RICAllo vs. Auto/(Auto) Median Months (m) or Percent at Year (y)
Garban et al. 2006 [50] Moreau et al. 2008 [51]	Flu 25 mg/m^2^ × 5 BU 2 mg/kg × 2 ATG 2.5 mg/kg × 5	65 vs. 219 Del13 + β2-micro > 3 mg/L	11% vs. nr	55% at 3 y for all patients	Median 19 vs. 22 m	Median 34 vs. 49 m
Krishnan et al. 2011 [45] Giralt et al 2019 [46]	TBI 2 Gy	37 vs. 48 Del13 or β2-micro ≥ 4 mg/L	22% vs. 11% 6 y 22% vs. 11% 10 y	47% vs. 77% 6 y 57% vs. 86% 10 y *p* = 0.005 6 y *p* = 0.004 10 y	31% vs. 13% 6 y 21% vs. 4% 10 y	51% vs. 47% 6 y 37% vs. 29% 10 y
Björkstrand et al. 2011 [47] Gahrton et al. 2013 [48]	Flu 30 mg/m^2^ × 3 TBI 2 Gy	29 vs. 63 Del13	NR	NR	31% vs. 10% 5 y 21% vs. 5% 8 y *p* = 0.016 5 y *p* = 0.026 8 y	69% vs. 52% 5 y 47% vs. 31% 8 y
Knop et al. 2019 [52]	Melphalan 140 mg/m^2^ Fludarabine 30 mg/m^2^ × 3	126 vs. 73 Del13 Del13 + del17p	14% vs. 4% 2 y *p* = 0.008 NR	NR NR	Median 34.5 vs. 21.8 m *p* = 0.003 37.5 vs. 6.1 m *p* = 0.0002	Median 70.2 vs. 71.8 m 61.5 vs. 23.4 m *p* = 0.032
Costa L 2020 [49] Pooled data from Italian PETHEMA EBMT BMT-CTN	TBI 2 Gy and other	89 vs. 125 Del13 or β2 micro ≥ 4 mg/L	NR	NR	32% vs. 17% 5 y 22% vs. 9% 10 y *p* = 0.015 5 y *p* = 0.008 10 y	52% vs. 51% 5 y 39% vs. 29% 10 y

Bu: busulfan; ATG: antithymocyte globulin.

**Table 3 jcm-09-02180-t003:** Studies from 2017 of allogeneic transplantation in relapsed and/or refractory multiple myeloma patients (RRMM).

Reference	Number of Patients	Conditioning and Type of Transplant	NRM Percent (at Year) (y) or Month (m)	Relapse Rate Percent (at Year) (y) or Month (m)	PFS Median Months (m) or Percent at Year (y)	OS Median Months (m) or Percent at Year (y)
Castagna et al. 2017 [60]	*n* = 30	Heavily pretreated Haplo Post-transplant Cyclophosphamide	10% at 18 m	42% at 18 m	33% at 18 m	63% at 18 m
Schneidawind et al. 2017 [61]	*n* = 41 Relapse *n* = 11 Refractory *n* = 30	Myeloablative or Reduced intensity Post-Allo maintenance *n* = 20	20% at 3 y	65% at 3 y	15% at 3 y	51% at 3 y 68% at 3 y
Sobh et al. 2017 [62]	10/10 MUD *n* = 419 9/10 MMUD *n* = 93 CB *n* = 58	Allo/RIC various conditioning	MUD 22% 2 y MMUD 33% 2 y CB 27% 2 y	MUD 58% 5 y MMUD 37% 5 y CB 69% 5 y	MUD 14% 5 y MMUD 27% 5 y CB 4% 5 y	MUD 33% 5 y MMUD 39% 5 y CB 25% 5 y
Patriarca et al. 2018 [63]	*n* = 169 Don *n* = 79 No Don *n* = 90	RL after first Auto AlloRIC Bortezomib + IMiD	27% 5 y	NR	18% 7 y 0% 7 y *p* = 0.0001	31% 7 y 9% 7 y *p* = 0.0001
Kawamura et al. 2018 [64]	*n* = 65	Heavily pretreated 1–7 lines + Auto	23.4% 3 y	57.8% 3 y	18.8% 3 y	47.2% 3 y
Ikeda et al. 2019 [65]	Allo *n* = 192 ReAuto *n* = 334 Multicenter Registry study	Myeloablative *n* = 38 RIC *n* = 153	NR	NR	NR	23.8% 5 y 33,7% 5 y
Greil et al. 2019 [66]	*n* = 109 Salvage + first line) Salvage *n* = 63 First line *n* = 46	Prior Auto *n* = 96 RIC/Flu conditioning Heavily pretreat High risk	12.4% 10 y 13.4% 2 y 10% 2 y	67.6% 10 y 71.4% 2 y 24.8% 2 y	20.1% 10 y 3.7% 5 y 46.5% 10 y	26.1% 10 y 23.7% 5 y 50.2% 10 y
Sahebi et al. 2019 [67]	*n* = 96	Haplo	21% 1 y	56% 2 y	17% 2 y	48% 2 y
Bryant et al. 2020 [46]	*n* = 73	CD 34 selected No GVHD prophylaxis	22% 1 y	47% 3 y	30% 3 y	50% 3 y

NRM, nonrelapse mortality; IMiDs, immunomodulatory drugs; MUD, matched unrelated donor; MMUD, mismatched unrelated donor.

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
