# Peer review of "Allogeneic Transplantation in Multiple Myeloma—Does It Still Have a Place?"

_jcm, 2020, doi:10.3390/jcm9072180_

Round 1

Reviewer 1 Report

Gahrton et al, present a very well written overview of allogeneic stem transplantation for multiple myeloma. It is timely in drawing attention to the efficacy of this modality in the current era, where novel drug therapy has supplanted the use of this classical methods of treating hematological malignancy potentially depriving patients of the benefit to be derived from it. This is particularly so since in a number of patients durable benefit of an allograft may be seen with timely application, and that benefit is being denied patients due to the narrative which has surrounded novel therapy, which while effective, is seldom so in the long term. To being this into sharper relief, I would suggest that the authors should consider revising some aspects of the article to improve the focus on contemporary management of MM patients. 

  1. The emphasis on MAC at the outset of the article distracts from the contemporary widely adopted and utilized allograft modality, RIC. I would suggest mentioning it in a brief historical and concept building context and replace this section this with a later discussion of the authors' expert opinion on interfacing allogeneic transplantation in the management schema of high risk and relapsed patients. This should be contextualized with the more modern auto transplant data coming from studies such as the IFM-DFCI trial comparing novel induction and maintenance with auto transplant. 
  2. Redundancies between the text and tables for transplant trials comparing auto vs allo SCT should be minimized to concisely deliver the authors' opinion on the value of upfront of allo SCT. This is no longer a widely accepted therapy in standard risk MM because of the upfront TRM risk and availability of very effective non-transplant therapy. Therefore this too should be abbreviated to emphasize the discussion of this strategy in the management of high risk patients. 
  3. In keeping with the title of the article, 'does it still have a place', the sections on both high risk myeloma and relapse should contain a discussion of the relative benefit of allogeneic SCT over contemporary approaches utilized in these settings, such as the use of MoAb and second line PI containing regimens. Highlighting the value of the GVM effect off allografting and DLI in these subsets of patients over the limited time efficacy of some of the contemporary treatment modalities will be helpful.     
  4. The section on CAR T/NK cells and NK cells seems to be somewhat forced into this paper and not consistent with the title and subject being covered and should be removed. The space gained should be utilized for the authors' expert discussion of appropriate timing of allografts along with contemporary therapies. In this reviewer's opinion this is particularly important in view of the prevailing confusion in the field on how best to sequence the many available effective therapies, in a disease which occupies a wide continuum of risk categories. This has created a major reluctance in the field on the use of allografting to treat myeloma, depriving patients of the benefits of this effective therapy. Addressing some of these questions, beyond stating the trial outcome results in this review will be very helpful. 

Author Response

Response to Reviewer 1

  1. In accordance with the reviewer the section on MAC has been reduced and revised (red in last paragraph lines 82-86), but not completely deleted since it is the background for the later reduced intensity conditioning studies. Also, it clearly shows that higher intensity reduces relapse rate. As suggested by the reviewer Allo has been positioned in the last rewritten section (in red) and related to the later results with new drugs and Auto.
  2. Redundancies have been reduced and focus has been on clarifying the authors view on using allo upfront in very special cases. Revision is seen in red lines 126 – 128; 144-151; 159-161; 168 - 171
  3. This is a very important comment Allo has now been positioned and related to other treatments in the last rewritten section
  4. I agree that this section may be somewhat out of scope. This section has now been deleted and replaced, as suggested by the reviewer by a section positioning Allogeneic transplantation and related to other treatments. Just a remaining comment on a possible use in combination with CAR-T and CAR NK

Reviewer 2 Report

This review paper focused on the current status of allogeneic stem cell transplantation (Allo) in patients with multiple myeloma. The authors summarized the results of the recent clinical trials as a part of front line therapy and salvage therapy. They recommended reduced-intensity conditioning Allo (RICAllo) following autologous stem cell transplantation as upfront treatment especially in younger patients with poor-risk cytogenetics. They also discussed the role of consolidation and maintenance therapy with novel agents and the potential of Allo in combination with CAR-T cell therapy.

Minor comments:

  1. Although the authors described the comments on chromosomal abnormalities and plasma cell leukemia in text, it might be better to describe these after the section of “upfront transplantation of high-risk patients”.

Author Response

Response to Reviewer 2.

  1. The reviewer suggest that Plasma cell leukemia is moved to the section upfront transplantation in high risk patients. This is a very valid proposal. However, since we discuss only prospective trials in this section, and no such studies are available in plasma cell leukemia, we would prefer to keep it separate. Also, we think that plasma cell leukemia is a very special disease and therefore worth a separate section. Thus, we would prefer not to move this section.

Round 2

Reviewer 1 Report

Thank you. The authors have addressed my comments

This manuscript is a resubmission of an earlier submission. The following is a list of the peer review reports and author responses from that submission.